# Brief communication: Real-time estimation of optimal tip-speed ratio for controlling wind turbines with degraded blades

Devesh Kumar[1,2] and Mario A. Rotea[3]

[1]Center for Wind Energy and Department of Electrical and Computer Engineering, The University of Texas at Dallas, Richardson, TX 75080 USA.
[2]Now with Drive System Design Inc., Farmington Hills, MI 48335, USA
[3]Center for Wind Energy and Department of Mechanical Engineering, The University of Texas at Dallas, Richardson, TX 75080 USA.

**Correspondence:** Mario A. Rotea (rotea@utdallas.edu)

**Abstract.** Rotor performance is adversely affected by wear and tear of blade surfaces caused, for example, by rain, snow, icing, dirt, bugs, ageing, etc. Blade surface degradation changes the aerodynamic properties of the rotor, which in turn changes the optimal tip-speed ratio (TSR) and the corresponding maximum power coefficient. Below rated wind speed, if a turbine continues to operate at the manufacturer designed optimal TSR, the rotor power could
decrease more than necessary unless the optimal TSR is corrected to compensate for blade degradation or blade surfaces are restored. Re-tuning the tip-speed ratio can lead to an improvement in energy capture without blade repairs. In this work, we describe a real-time algorithm to re-tune the tip-speed ratio to its optimal, but unknown, value under blade degradation. The algorithm uses power measurements only and a Log-of-Power Proportional-Integral Extremum Seeking Control (LP-PIESC) strategy to re-tune the TSR. The algorithm is demonstrated in
simulations to command the TSR set-point required by a generator speed control loop that maximizes power below rated wind speeds. Comparison of this solution with a baseline controller that uses the optimal TSR for a rotor with clean blades demonstrates improvements in energy capture between 0.5% and 3.4%, depending on the severity of blade degradation and the wind conditions. These results are obtained using the OpenFAST simulation tool, the NREL 5-MW reference turbine, and the Reference Open-Source Controller developed by the US National Renewable
Energy Laboratory.

## 1   Introduction

Below-rated wind speed, a wind turbine is typically controlled to maximize the power extracted from the wind. In this regime, the rotor power is proportional to the power available in the wind times the power coefficient ($C_P$). For a typical variable-speed variable-pitch wind turbine, $C_P$ is a unimodal function of the tip-speed ratio ($\lambda$) and the
blade-pitch angle ($\beta$) (Manwell et al., 2010; Burton et al., 2011). This implies that there is an optimal tip-speed ratio and blade-pitch angle for maximizing $C_P$ and hence the output power. Intuitively, maximizing power below-rated

wind speeds requires keeping the blade-pitch angle constant at its ideal value $\beta_{\mathrm{opt}}$ while adjusting the rotor speed to maintain the optimal tip-speed ratio $\lambda_{\mathrm{opt}}$ despite wind speed changes (Pao and Johnson, 2011; Burton et al., 2011; Abbas et al., 2022).

Rotor performance is adversely affected by wear and tear of blade surfaces caused, for example, by rain, snow, icing, dirt, bugs, ageing, etc. Blade surface degradation changes the aerodynamic properties of the rotor, which in turn changes the optimal tip-speed ratio $\lambda_{\mathrm{opt}}$ and the corresponding maximum power coefficient $C_{\mathrm{P}}^{\mathrm{max}}$. If a turbine continues to operate at the originally designed $\lambda_{\mathrm{opt}}$, the rotor power can decrease more than necessary unless the optimal tip speed ratio is corrected to compensate for blade degradation. Re-tuning the optimal tip speed ratio in

these off-design conditions can lead to an improvement in energy capture.

Annual Energy Production (AEP) losses due to blade degradation have been reported in the literature. For example, Han et al. (2018) showed that contamination and erosion at leading edge of blade tips can reduce AEP by 2%-3.7%. Ehrmann et al. (2013, 2017); Wilcox et al. (2017) studied the effect of surface roughness on wind turbine performance. Ehrmann et al. (2013) observed that roughness leads to a consistent increase in drag compared to a

clean configuration. Ehrmann et al. (2017) showed that the maximum lift-to-drag ratio decreases 40% for $140\mu m$ roughness, corresponding to a 2.3% loss in AEP, approximately. AEP losses of 4.9% and 6.8% for a NACA $633-418$ and an NREL S814 airfoils, respectively, operating with $200\mu m$ roughness were observed in Wilcox et al. (2017). Wilcox and White (2016) studied the power loss due to insect contamination on the blades. They concluded that insect impingement simulations should be considered in airfoil design. A numerical approach capable of simulating

the ice accretion transient phenomenon and its effects on wind turbine performance was presented in Zanon et al. (2018). This reference shows that keeping the tip-speed ratio at its designed optimal value can reduce the power coefficient by 3% after the icing event.

In this brief communication we attempt to answer the following question: Can re-tuning control parameters recover power/AEP losses before blade repairs are made? This question is considered in the context of torque control systems

that use optimal values of TSR to calculate the set-point for the generator angular speed that maximizes power.

In principle, on-line methods to estimate the power coefficient can be useful to answer this question. Due to space limitations, we do not elaborate on these methods. Instead, we provide references to such literature. See, for example, the work of Odgaard et al. (2008); De Kooning et al. (2013); Petković et al. (2013).

In this work, we explore the use of a recently develop variant of Extremum Seeking Control (ESC) for re-tuning

control parameters to maximize power capture for rotors with degraded aerodynamic performance. More specifically, we apply the Log-of-Power Proportional-Integral Extremum Seeking Control (LP-PIESC) algorithm to identify the optimal tip-speed ratio (TSR) for contaminated or eroded blades in real time. This algorithm requires one measurement only; i.e., the rotor power. The LP-PIESC has been shown to be a faster variant of the traditional perturbation-based ESC (Kumar and Rotea, 2022).

The LP-PIESC algorithm is used to identify optimal TSRs set points to be tracked by a generator speed control loop operating below-rated wind speeds. Due to its widespread availability, we have chosen the Reference Open-Source

Controller (ROSCO) developed by Abbas et al. (2022) and the NREL 5-MW reference turbine model (Jonkman et al., 2009) to illustrate our approach.

The use of extremum seeking control to identify optimal control parameters for a single wind turbine is not new, an early reference is Creaby et al. (2009), followed by work using large eddy simulations in Ciri et al. (2017) and an experimental campaign in Xiao et al. (2019). Due to the lack of consistent convergence across different wind speeds of the standard ESC, the log-of-power extremum seeking (LP-ESC) was introduced in Rotea (2017) to have predictable consistent convergence time and to produce an algorithm that after calibration at one single wind speed exhibits the same performance at all below-rated wind speeds. The LP-ESC was then tested using high-fidelity simulations to demonstrate its advantages over the conventional ESC in Ciri et al. (2019).

The organization of the paper is as follows. The main characteristics of ROSCO are given in Section 2. This section also describes the specific blade degradation cases considered (contamination and erosion) as well as a brief description of the LP-PIESC algorithm. This algorithm has two distinct (but coupled) functions. The control function, which is described in Section 2 and a parameter estimation function, whose main features are given in Appendix A. Section 3 focuses on the real-time identification of optimal tip-speed ratio with LP-PIESC. To facilitate the manuscript's readability, the algorithm parameters and the most relevant equations for parameter estimation are provided in Appendix A. The results of simulations using OpenFAST (NREL, 2020, accessed: February 12, 2024) for several wind profiles are given in Section 3. These results provide numerical evidence that the LP-PIESC can find the unknown optimal tip-speed ratio despite variations in mean wind speed, turbulence intensity, and the level of blade degradation, thus increasing energy capture in off-design conditions. Conclusions are given in Section 4.

## 2  Background

### 2.1  Reference Open-Source Controller (ROSCO)

The Reference Open-Source Controller (ROSCO) has been introduced in Abbas et al. (2022) to update the legacy NREL 5-MW controller (Jonkman et al., 2009). The ROSCO is available for download and implementation on GitHub: https://github.com/NREL/ROSCO (accessed on February 12, 2024). This controller is proposed as a modern control architecture that can be deployed on multiple wind turbine models and simplifies the tuning procedure in OpenFAST (NREL, 2020, accessed: February 12, 2024). In this architecture, both the generator torque ($\tau_\mathrm{g}$) and the blade pitch angle ($\beta$) are governed by PI controllers. The set-point tip-speed ratio ($\lambda_{sp}$) is a tunable parameter required to determine the generator speed reference value. Using the set-point tip-speed ratio and estimated wind speed ($\hat{v}$), a generator speed reference ($\omega_\mathrm{g,ref}$) is obtained from Eq. (1).

$$\omega_\mathrm{g,ref} = \frac{\lambda_{sp}\hat{v}N}{R} \tag{1}$$

where $R$ is the rotor radius, and $N$ is the gear ratio. The generator speed reference $\omega_{\mathrm{g,ref}}$ is the reference input to the generator torque PI controller for generator speed ($\omega_{\mathrm{g}}$) control to maximize the power capture in below-rated wind speed. Details of the ROSCO framework can be found in Abbas et al. (2022).

In this work, we tune the set-point tip-speed ratio $\lambda_{sp}$ in the ROSCO controller using the LP-PIESC to maximize the power capture below-rated wind speed. We did not modify the control logic of the ROSCO and used it as it is defined for the NREL 5-MW reference turbine in OpenFAST. In our simulations, ROSCO takes wind speed estimate $\hat{v}$ from the rotor disk average (RtVAvgxh) calculated by OpenFAST. We do not use the wind speed estimator in the ROSCO because this estimator makes use of parameters that could change with blade degradation or contamination.

Recent methods with potential to provide accurate estimates of the wind speed despite changes in blade properties are in Lio et al. (2021), which requires real-time turbine time series to develop a wind speed estimator using a Gaussian process regression approach. A more expensive alternative would be using a LIDAR as done in Meng et al. (2022). We have not pursued any of these methods because our study is meant to provide evidence of the potential LP-PIESC has to estimate unknown parameters such as optimal TSR despite uncertainty in turbine parameters; see

also Kumar and Rotea (2022), where LP-PIESC is used to tune the torque gain and blade pitch angle to optimal values, starting from non-optimal parameters.

## 2.2    Performance loss due to leading-edge blade degradation

Rotor performance is adversely affected by wear and tear of blade surfaces caused, for example, by rain, snow, icing, dirt, bugs, ageing, etc. Blade surface degradation changes the aerodynamic properties of the rotor, which in turns

changes the manufacturer designed optimal tip-speed ratio ($\lambda_{\mathrm{opt}}$) and the corresponding maximum power coefficient ($C_{\mathrm{P}}^{\mathrm{max}}$). If a turbine continues to operate at the designed $\lambda_{\mathrm{opt}}$, the rotor power can decrease more than necessary unless the optimal tip speed ratio (TSR) is corrected to compensate for blade degradation. Re-tuning the tip-speed ratio in these off-design conditions can lead to an improvement in energy capture. In this article we provide numerical evidence demonstrating that extremum seeking control can re-tune the TSR to its new optimal value.

Han et al. (2018) studied the impact on annual energy production of blade leading edge contamination and erosion. In particular, they studied the aerodynamic performance of the blade tip airfoil (NACA64) for the NREL 5-MW wind turbine (Jonkman et al., 2009). To demonstrate the ability of LP-PIESC for finding the optimal TSR for rotors with degraded blades, we selected two cases from Han et al. (2018): (i) Contamination of blades - decreased the lift coefficient ($C_{\mathrm{l}}$) by 30% and increased the drag coefficient ($C_{\mathrm{d}}$) by 150% and (ii) Erosion of blades - decreased

the lift coefficient ($C_{\mathrm{l}}$) by 50% and increased the drag coefficient ($C_{\mathrm{d}}$) by 300%. These changes in lift and drag coefficients occur at angles of attack (AoA) between $-5°$ to $16°$. Note that the NACA64 airfoil is in the tip region of the NREL 5-MW turbine blade, approximately 30% of the blade length. A schematic with the degraded blade sector is in Figure 1. Lift and drag coefficients for the clean blade as well as the contaminated and eroded blades are shown in Figure 2.

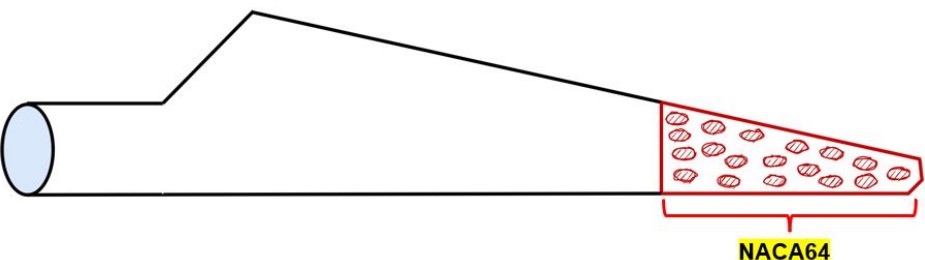

**Figure 1.** Tip section of the NREL 5-MW blade with NACA64 airfoil. It is 30% of the blade length, 43.05 m from the blade root to 61.5 m.

The $C_P - \lambda$ curves with degraded blades were obtained from NREL OpenFAST (NREL, 2020, accessed: February 12, 2024) using the modified lift and drag coefficients. Figure 3 shows the results for all the cases. Note that the designed optimal values for a clean blade are $\lambda_{\mathrm{opt}} = 7.6$ and $C_P^{\mathrm{max}} = 0.483$. The modification of the lift and drag coefficients leads to a shift in the optimal $C_P - \lambda$ curve. When the blade is contaminated the optimal TSR increases to $\lambda_{\mathrm{opt}} = 8.2$, with $C_P^{\mathrm{max}} = 0.431$. For the eroded blade, the curve shift is more pronounced and the maximum power coefficient drops to $C_P^{\mathrm{max}} = 0.351$ at $\lambda_{\mathrm{opt}} = 8.4$. We can observe from these plots that if the turbine is still controlled using the clean blade set-point $\lambda_{sp} = \lambda_{\mathrm{opt}} = 7.6$ for both the contaminated and degraded blade, it will produce less power than the maximum power it could produce should the set-point TSR were changed to their new optimal values. The $C_P$ loss would be roughly 1.4% for the contaminated blade and 3.5% for the eroded blade if the TSRs are not re-tuned. Thus, it is advantageous to change the set-point TSR to their new optimal values. In this study, the LP-PIESC is used to search for these new optimal tip-speed ratios.

### 2.3 Log-of-Power Proportional-Integral Extremum Seeking Control (LP-PIESC)

In this study, we use the Log-of-Power Proportional-Integral Extremum Seeking Control (LP-PIESC) strategy introduced in Kumar and Rotea (2022). The algorithm is gradient-based, which can adjust the tunable parameters to maximize a system's performance index in real-time without detailed physical models. The use of the Log-of-Power (LP) as feedback signal enables consistent convergence across varying mean wind speeds (Rotea, 2017; Ciri et al., 2019) and PIESC achieves rapid convergence to the optimum (Guay and Dochain, 2017).

Let us assume that we seek the value of the scalar parameter $u$ (tip speed ratio) that maximizes a scalar-valued function $f(u)$ (power). To solve this problem, the proportional-integral extremum seeking controller proposed in Guay and Dochain (2017) has been modified to add a back-calculation anti-windup and is given by (2):

$$\begin{cases} u = -k_p \hat{\theta}_1 + \hat{u} + d(t) \\ \dot{\hat{u}} = -\frac{1}{\tau_I} \hat{\theta}_1 + k_b(u^s - u) \\ u^s = \texttt{sat}(u) \end{cases} \tag{2}$$

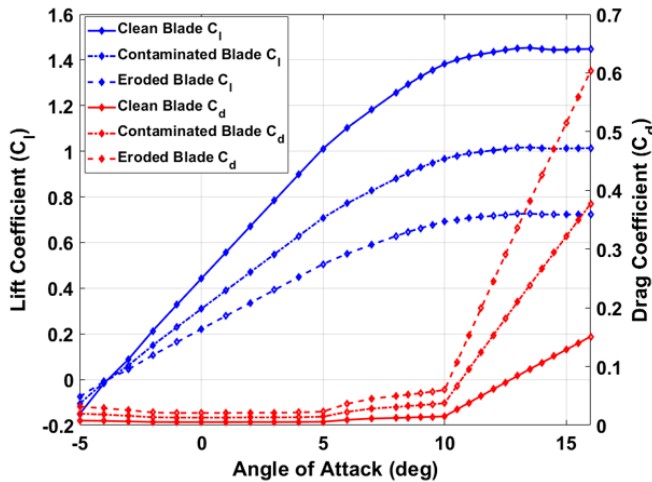

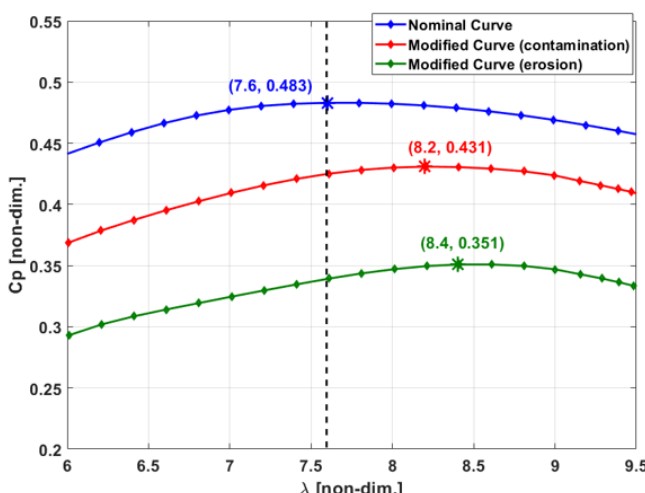

**Figure 2.** Change of lift and drag coefficients for NACA64 airfoil.

**Figure 3.** Nominal and modified (due to contamination and erosion of blades) $C_{\text{P}} - \lambda$ curve for NREL 5-MW wind turbine reference model.

where `sat`$(\cdot)$ is a linear function between TSR of 4 and 10, $k_p$ is the proportional gain, $\tau_I$ is the integral time constant, $k_b$ is the anti-windup gain and $\hat{\theta}_1$ is a scalar parameter to be estimated. This parameter is representative of the derivative of the log-of-power with respect to the change in tip speed ratio. A sinusoidal dither signal is chosen and is denoted by $d$. In comparison to the more popular version of extremum seeking (Krstić and Wang, 2000), which is essentially the integrator equation in (2) for $\hat{u}$, the proportional term $k_p\hat{\theta}_1$ accelerates convergence. The difference between the saturated output $u^{\text{s}}$ and the calculated controller output $u$ is fed back into the input of the integrator through an anti-windup gain $k_b$, which is similar to the design of anti-windup ESC proposed in Creaby et al. (2009). Intuitively, the addition of the proportional term increases the control bandwidth (speed of response) that results when substituting a pure integral controller with a PI control law.

The strategy utilized to find the unknown time-varying parameter $\hat{\theta}_1$ also contributes to convergence time improvements and is discussed in appendix A. This approach is not the conventional perturbation/demodulation method used to extract gradient information in previous versions of ESC (Krstić and Wang, 2000). Rather, it draws inspiration from continuous-time recursive least squares with forgetting as well as estimation of time-varying parameters and adaptive control (Guay and Dochain, 2017; Krstic et al., 1995; Shaferman et al., 2021).

## 3 Real-time identification of optimal TSR with LP-PIESC

As mentioned already, blade surface degradation changes the aerodynamic properties of the rotor, which results in shifting of the manufacturer designed optimal TSR. Therefore, real-time identification of the modified optimal TSR

for degraded blades is needed to avoid loss of energy. The LP-PIESC can be used to do online search of the modified optimal TSR in real-time.

## 3.1 LP-PIESC design

This subsection provides the basic block diagram of a turbine equipped with the LP-PIESC for optimal TSR estimation. The algorithm itself is described in detail in Eq. 2 and appendix A. The latter also contains a description of the tuning of parameters in the LP-PIESC.

### 3.1.1 System architecture

The NREL 5-MW turbine reference model with OpenFAST is used in the work (Jonkman, 2013). Table 1 lists the major parameters of this turbine model.

**Table 1.** Main parameters of NREL 5MW turbine.

| Description | Value |
|---|---|
| Rated Power | 5 MW |
| Rotor radius ($R$) | 63 m |
| Gear Ratio ($N$) | 97 |
| Cut-in, Rated, Cut-out wind speed | 3 m/s, 11.4 m/s, 25 m/s |
| Cut-in, Rated rotor speed | 6.9 rpm, 12.1 rpm |

This study considers a wind turbine operating below-rated wind speed, as this is the region where wind turbine control algorithms seek to maximize power. A high-level block diagram of the entire system is shown in Figure 4. The input to the LP-PIESC is the logarithm of the rotor power $P$ normalized with respect to the rated power $P_\mathrm{r} = 5$ MW after a moving average filter is applied to the instantaneous power signal to remove high-frequency fluctuations. The output of the LP-PIESC is the estimate of the set-point tip-speed ratio ($u^s$). A rate limit of 0.1/s is applied to the estimated set-point tip-speed ratio to produce the actual TSR set-point ($\lambda_{sp}$) for the ROSCO controller (as in Eq. (1))

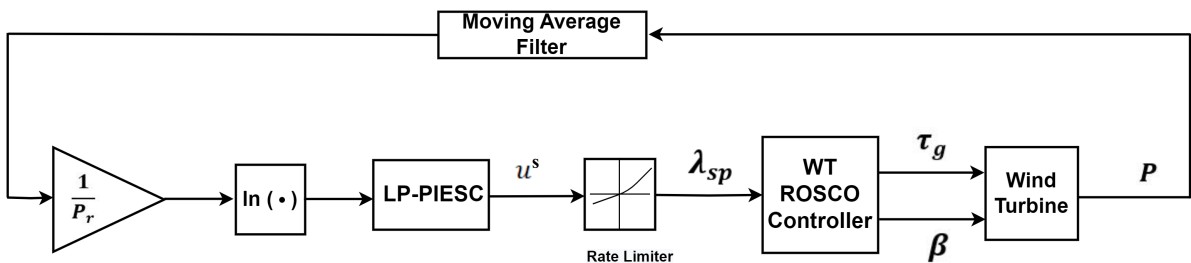

**Figure 4.** LP-PIESC implementation with ROSCO wind turbine controller.

The LP-PIESC parameters (Eq. 2 and appendix A) are designed using a clean blade (no contamination or erosion) and then fixed at these design values for implementation with the degraded blade cases. First the most relevant dynamics is identified using step responses to select dither frequency and amplitude. Then, the remaining parameters of the LP-PIESC are calibrated by trial and error to achieve convergence to the optimum TSR for the clean blade at 8 m/s wind speed and with 10% turbulence intensity. To maintain continuity in the exposition, the equations for the LP-PIESC parameter estimation methods and the numerical values for all the algorithm parameters can be found in appendix A.

## 3.2 Results

### 3.2.1 Simulation conditions

The LP-PIESC controller with the parameters shown in Appendix A, Table A1, is evaluated with OpenFAST simulations for hub-height mean wind speeds of 7 m/s, 8 m/s, and 9 m/s, vertical shear exponent $\alpha = 0.2$ and under turbulence intensities of 10% and 15%, respectively.

The wind profiles for the simulations were obtained using NREL TurbSim (Jonkman, 2009). TurbSim follows IEC 61400-1 (IEC, 2005) to generate the wind profiles. We specified five parameters in the TurbSim input file to generate the wind input files for our simulations: (1) Turbulence model is chosen as Kaimal (IECKAI), (2) IEC turbulence type is set as Normal Turbulence Model (NTM), (3) Hub-height is 90 m for NREL 5MW reference turbine, (4) Mean wind speed, and (5) Turbulence intensity in percentage. All other parameters were left to their default values in TurbSim. The mathematical expression for the Kaimal spectrum can be found in IEC (2019).

Figure 5 illustrates the time series of the hub-height wind speeds with means of 7 m/s, 8 m/s, and 9 m/s and 10% turbulence intensity (TI). We use these wind profiles to evaluate the performance of the LP-PIESC for both the contaminated blade case and the eroded blade case (from section 2.2). We also simulate cases with 15% TI and same mean values. In all simulations, we set the tip-speed ratio set point to the ROSCO at the clean blade optimum; i.e., $\lambda_{sp} = \lambda_{\mathrm{opt}} = 7.6$, for the first 500 s of the simulation and then turn on the LP-PIESC to evaluate the convergence of the TSR set point to the new optimal values for contaminated and eroded blades, respectively.

### 3.2.2 Simulation Results

Results for the contaminated blade with the wind profile in Figure 5 are shown in Figure 6. The top plot shows the tip-speed ratio set-point ($\lambda_{sp}$) given to the ROSCO (see Figure 4), the middle plot shows the actual tip-speed ratio ($\lambda$) output and the estimated power coefficient ($C_{\mathrm{P}}$) is shown in the bottom plot; these latter two parameters are from OpenFAST. Both the tip-speed ratio ($\lambda$) and the power coefficient ($C_{\mathrm{P}}$) time series are shown after applying a 100 s moving average filter to the OpenFAST outputs to smooth these signals. Recall from Figure 3 that the optimal value of the tip-speed ratio and the maximum value of $C_{\mathrm{P}}$ for this case are 8.2 and 0.431, respectively.

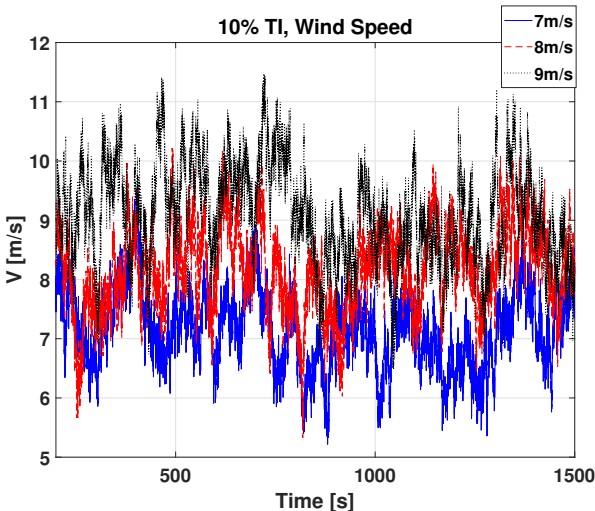

**Figure 5.** Wind speed time series at hub height: mean wind speeds 7m/s, 8m/s, 9m/s, and 10% turbulence intensity.

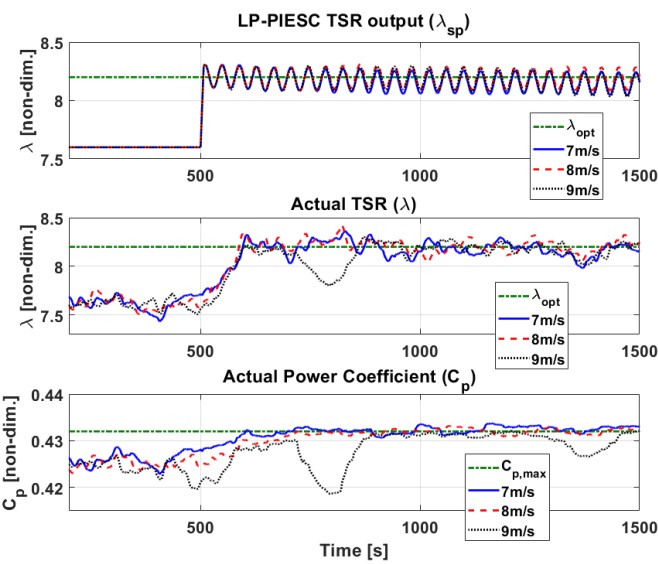

**Figure 6. Contaminated blade** - Performance of the LP-PIESC with the parameters in Table A1 and hub-height wind speed from Figure 5. The LP-PIESC is turned on at 500 s. Rate-limited LP-PIESC tip-speed ratio set-point $\lambda_{sp}$ (**top**). Tip speed ratio $\lambda$ (**middle**) and estimated power coefficient $C_{\mathrm{P}}$ (**bottom**) from OpenFAST output file. The dashed green horizontal lines indicate true optimal parameters $\lambda_{\mathrm{opt}}$, and $C_{\mathrm{P,max}}$ (Figure 3).

The LP-PIESC converges to the new optimal tip-speed ratio almost instantaneously for all the cases. The actual tip-speed ratio ($\lambda$) and the estimated power coefficient ($C_{\mathrm{P}}$) converge in less than 100 s. With 9 m/s mean wind speed

there are some drops in the estimated power-coefficient ($C_P$) around 500 s, 800 s and 1400 s. From Figure 5, we can see that during those instances the wind speed moves into above-rated operation of the NREL 5-MW turbine (Jonkman et al., 2009). This can also be seen from Figure 7 where the blade pitch (bottom right plot) is activated at those times to regulate the generator speed close to its rated value 1173.7 rpm. The increase to above-rated wind speed, and approximate regulation of generator speed provided by ROSCO, explain the dips observed in power coefficient and tip-speed ratio in Figure 6.

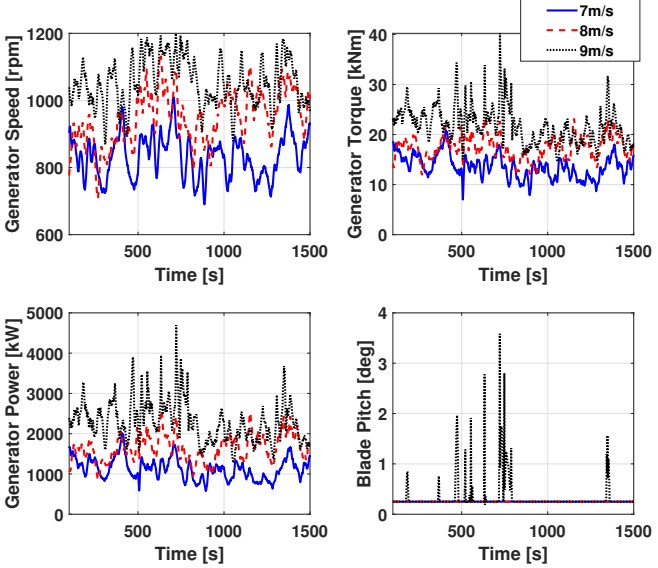

**Figure 7. Contaminated blade** - Response of turbine parameters with the LP-PIESC and the wind profiles in Figure 5.

We also evaluate the performance of the LP-PIESC with an increased turbulence intensity of 15%. The simulation results are shown in Figure 8. It is observed that increasing the turbulence intensity does not affect the transient or steady-state performance of the LP-PIESC. The LP-PIESC continues to converge quickly, and the drop in $C_P$ for 9 m/s wind speed is due to the turbine entering the above-rated wind speed region.

Next, we evaluate the performance of the LP-PIESC for the eroded blade case using the same mean wind speeds and turbulence intensity as before. Recall from Figure 3 that the optimal value of the tip-speed ratio and the maximum value of $C_P$ for this case are 8.4 and 0.351, respectively. Results for this case are shown in Figure 9 and 10. It can be observed that as the LP-PIESC is turned on at 500 s, it converges to the optimal tip-speed ratio almost instantaneously for all the cases. The drops in $C_P$ for 9 m/s mean wind speed can be explained as before.

The results in this section provided evidence, via simulations, that the LP-PIESC can quickly find the unknown optimal tip-speed ratio despite variations in mean wind speed, turbulence intensity, and the level of blade degradation.

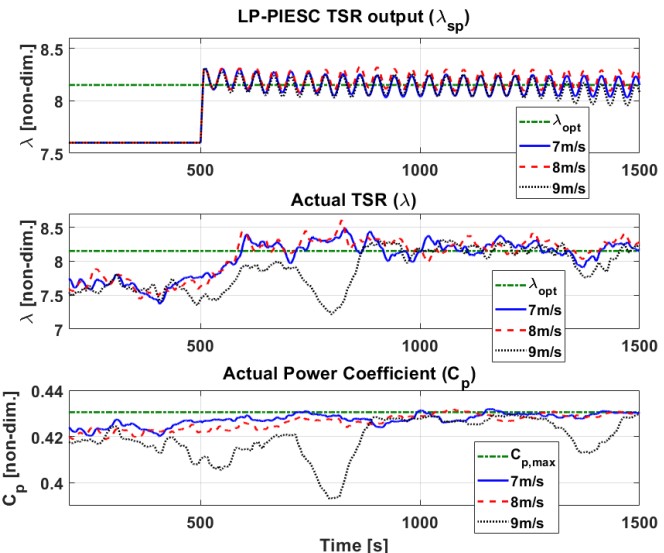

**Figure 8. Contaminated blade** - Performance of the LP-PIESC with the parameters shown in Table A1 and wind input with 15% TI. The LP-PIESC is turned on at 500 s. Rate-limited LP-PIESC tip-speed ratio set-point $\lambda_{sp}$ (**top**) , tip speed ratio from OpenFAST output $\lambda$ (**middle**), estimated power coefficient $C_P$ (**bottom**). The dashed green horizontal lines indicate true optimal parameters $\lambda_{opt}$, and $C_{P,max}$ (Figure 3).

### 3.2.3 Energy Capture

The energy capture using the LP-PIESC for both the contaminated blade case and the eroded blade case (Figure 3) is evaluated and compared with the baseline controller (i.e., the controller with the tip-speed ratio constant and corresponding to clean blades). The controllers are evaluated for hub-height mean wind speeds of 7 m/s, 8 m/s, and 9 m/s, vertical shear exponent $\alpha$=0.2 and turbulence intensities of 10% and 15%, respectively. TurbSim (Jonkman, 2009) is used to generate the wind profiles from six different seeds for each wind speed and turbulence intensity. An 230 average energy capture over those six wind profiles is presented here. All the calculations are done using the data from the time LP-PIESC is turned on (500 s) till the end of the simulation (1500 s).

The contaminated blade case is shown in Figure 11. The average energy capture with the LP-PIESC is compared with that of the baseline controller. Both the controllers were applied to the same blades (contaminated for this case). The baseline controller applies a constant tip-speed ratio set-point $\lambda_{sp} = 7.6$ to ROSCO while the LP-PIESC 235 applies the optimized tip-speed ratio time series $\lambda_{sp}$ from Figures 6 and 8. The energy capture with the LP-PIESC for the eroded blade was also compared with the baseline controller using the same approach. The average energy capture comparison for the eroded blade is shown in Figure 12.

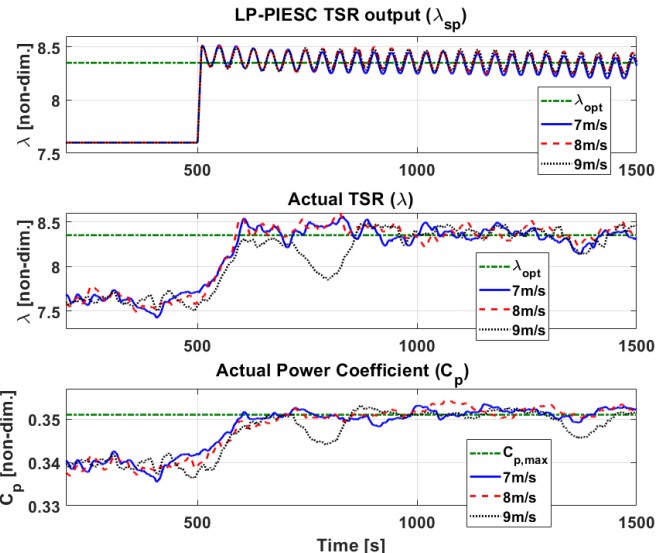

**Figure 9. Eroded blade** - Performance of the LP-PIESC with the parameters shown in Table A1 and hub-height wind from Figure 5. The LP-PIESC is turned on at 500 s. Rate-limited LP-PIESC tip-speed ratio set-point $\lambda_{sp}$ (**top**). Tip speed ratio $\lambda$ (**middle**) and estimated power coefficient $C_\mathrm{P}$ (**bottom**) from OpenFAST output file. The dashed green horizontal lines indicate true optimal parameters $\lambda_\mathrm{opt}$, and $C_\mathrm{P,max}$ (Figure 3).

The results suggest that the LP-PIESC can find the unknown optimal tip-speed ratio for degraded rotor blades and improve the energy capture. The maximum improvement is observed with the mean wind speed of 7 m/s, 10%
TI wind for both the contaminated (1.5%) and the eroded blade (3.4%) cases. It is interesting to note that these improvements in energy capture are very close to the reported improvements in the power coefficient in Section 2.2 as explained in the caption of Figure 3. The percentage energy increase for the mean wind speed of 9 m/s was the lowest. In this case, turbulence takes the wind speed and generator speed in the above-rated region where the turbine blade-pitch controller gets activated to constrain the generated speed to the rated value.

**4   Conclusions**

A log-power feedback PIESC (LP-PIESC) algorithm is presented to estimate the optimal tip speed ratio (TSR) below rated wind speeds despite changes in the rotor aerodynamics. Knowledge of the optimal TSR is necessary when the wind turbine controller uses this information to determine to the generator speed set point for optimal power extraction.
The LP-PIESC ability to identify optimal TSR set points is demonstrated using OpenFAST simulations with the the ROSCO reference controller introduced in Abbas et al. (2022). Optimal TSRs for blades with contaminated or

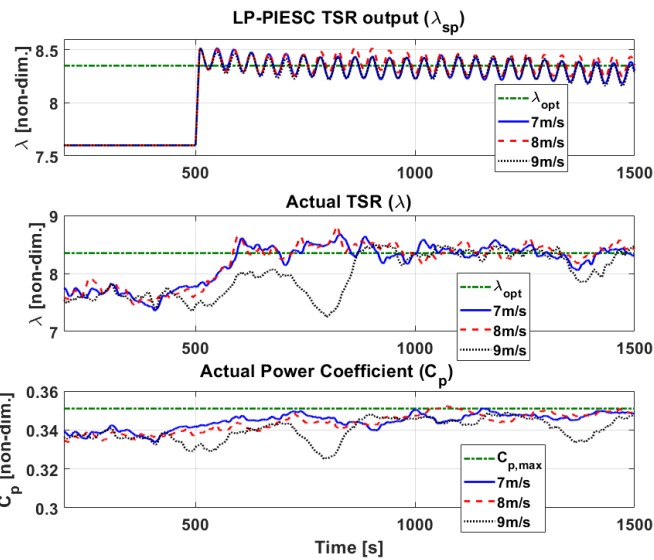

**Figure 10. Eroded blade** - Performance of the LP-PIESC with the parameters shown in Table A1 and wind input with 15% TI. The LP-PIESC is turned on at 500 s. Rate-limited LP-PIESC tip-speed ratio set-point $\lambda_{sp}$ (**top**) , tip speed ratio from OpenFAST output $\lambda$ (**middle**), estimated power coefficient $C_P$ (**bottom**). The dashed green horizontal lines indicate true optimal parameters $\lambda_{opt}$, and $C_{P,max}$ (Figure 3).

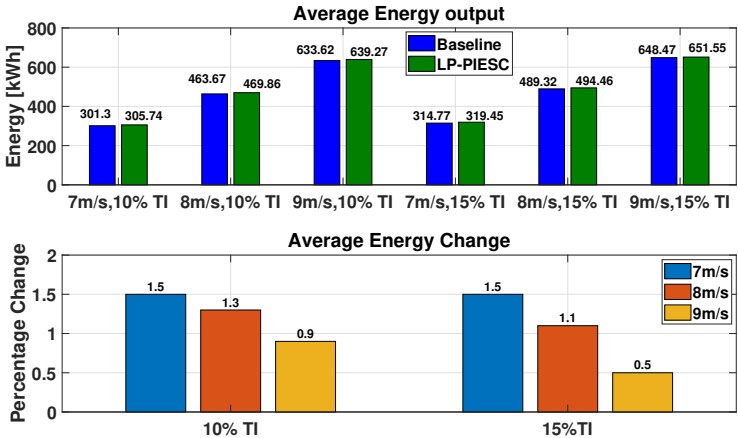

**Figure 11. Contaminated blade** - Energy capture comparison: Baseline vs. LP-PIESC. Average energy output with clean-blade optimal tip-speed ratio and LP-PIESC (**top**), percentage change in energy capture (**bottom**).

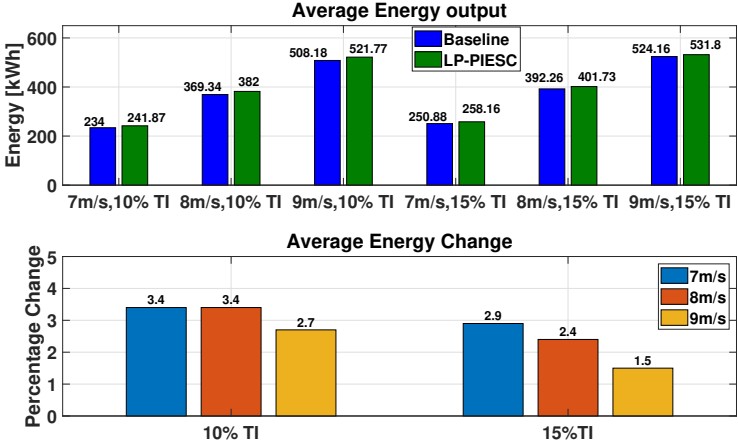

**Figure 12. Eroded blade** - Energy capture comparison: Baseline vs. LP-PIESC. Average energy output with clean-blade optimal tip-speed ratio and LP-PIESC (**top**), percentage change in energy capture (**bottom**).

eroded airfoils can be found despite variations in mean wind speed or turbulence intensity. The simulations show that re-tuning the TSR to optimal values can lead to increases in energy capture ranging from 0.5% to 1.5 % for contaminated blades and from 1.5% to 3.4% for eroded blades. The highest energy increases occur at lower wind
speeds, which is a favorable situation. Energy increases with eroded blades are higher than with contaminated blades for all cases considered. These positive results need to be balanced with the fact that for both contaminated and eroded blades optimal TSRs increase, which requires higher rotor speeds. The implications of this fact on the progression of blade degradation would need to be better understood. However, the method proposed can be used to increase energy capture until blades need to be cleaned or repaired.

The LP-PIESC technique provides rapid and consistent convergence across different below-rated wind speed conditions. Calibration of algorithm parameters at one wind condition works at other wind conditions. However, the design of the PIESC algorithm requires tuning more parameters than the conventional ESC. Therefore, additional work is needed to establish practical guidelines for parameter design. The dither signal does create a harmonic component in the generator torque. While the impact of this dither-induced harmonic has not been studied, it
should be noted that a stopping criterion could be added to eliminate the dither or the dither could be turned off after a fixed number of cycles. In fact, given the speed of convergence, even a partial cycle may suffice to estimate the optimal TSR. It is also important to note that if the turbine controller requires the $C_{\mathrm{P}} - \lambda$ curve for wind speed estimation or another control function, then estimating the optimal TSR set-point under blade degradation may not suffice to realize power gains; additional parameters would need to be estimated in this case to optimize power for
aerodynamically degraded rotors.

*Code availability.* Not Applicable

*Data availability.* The data presented in this work can be made available upon request.

## Appendix A: Algorithm and design parameters for LP-PIESC estimation

The method utilized to determine the unknown time-varying parameter $\hat{\theta}_1$ contributes to the improvement in con-
vergence time for this class of extremum-seeking control algorithms. This appendix describes the main features of
the parameter estimation used. Let $y(t)$ represent the log-power signal entering the LP-PIESC algorithm shown in
Fig. 4. Using Guay and Dochain (2017), the rate of change of $y$ is parameterized as

$$\dot{y} = \theta_0 + \theta_1(u - \hat{u}) = \phi^T\theta \tag{A1}$$

where $u$ and $\hat{u}$ are defined in Eq. (2), $\phi = [1, (u - \hat{u})]^T$ is the "regressor" in Eq. (A1) and $\theta = [\theta_0, \theta_1]^T$ the two unknown
time varying parameters. Although $\theta_0$ is not used in the control equation Eq. 2, this parameter is required to estimate
$\theta_1$ properly (Guay and Dochain, 2017).

The parameter vector $\theta$ is estimated by minimizing an output prediction error $e$ of the log of power signal $y$. The
prediction error $e$ is computed using

$$e = y - \hat{y} \tag{A2}$$

where $\hat{y}$ is the prediction of the output $y$, obtained from the following ODE

$$\dot{\hat{y}} = \phi^T\hat{\theta} + Ke + c^T\dot{\hat{\theta}}. \tag{A3}$$

The positive scalar $K$ is a parameter to be determined, and $c(t)$ is a filtered regressor calculated from

$$\dot{c}^T = -Kc^T + \phi^T \tag{A4}$$

It should be noted that the output prediction dynamics $\dot{\hat{y}}$ in (A3) comprises of two additional terms, one propor-
tional to the error $e$ and the other to the time derivative of the estimated time-varying parameter $\dot{\hat{\theta}}$, in addition to
a model of the dynamics $\dot{y}$. Intuitively, the addition of this latter term facilitates tracking time varying parameters.

Finally, The updating law for parameter estimation is

$$\dot{\hat{\theta}} = \text{Proj}(\Sigma^{-1}(c(e - \hat{\eta}) - \sigma\hat{\theta}), \hat{\theta}) \tag{A5}$$

where the Lipschitz projection operator $\text{Proj}(\cdot)$ is used to assure stability and that the estimates are bounded within
the constraint set. This projection algorithm was implemented as described in Appendix E of Krstic et al. (1995)

and the constraint set adaptation was adopted in accordance with Adetola and Guay (2011). The auxiliary variable estimate $\hat{\eta}$ is given by

$$\dot{\hat{\eta}} = -K\hat{\eta} \tag{A6}$$

The $2 \times 2$ matrix $\Sigma$ is the solution of the matrix differential equation

$$\dot{\Sigma} = cc^T - k_T\Sigma + \sigma I \tag{A7}$$

with $\sigma$ and $k_T$ as positive scalar constants. The inverse of $\Sigma$ is given by the solution of the ODE

$$\dot{\Sigma}^{-1} = -\Sigma^{-1}cc^T\Sigma^{-1} + k_T\Sigma^{-1} - \sigma\Sigma^{-2} \tag{A8}$$

This is the ODE we integrate in the algorithm to obtain the gain matrix in the update law (A5). The matrix update law in Eq. (A7) is similar to the one in continuous-time least-squares with forgetting Shaferman et al. (2021).

A condition for the convergence of the PI-ESC algorithm (Guay and Dochain, 2017) is the level of excitation provided by the filtered regressor $c(t)$ in (A4). This is quantified by the following persistence of excitation (PE) condition: there exists constants $\rho > 0$ and $T > 0$ such that

$$\int_t^{t+T} c(\tau)c(\tau)^T \, d\tau \geq \rho I \quad \forall t > 0 \tag{A9}$$

The dither signal provides a sufficient condition to satisfy (A9), which is the PE condition in assumption 5 of Guay and Dochain (2017). This assumption is used to prove the convergence of the PIESC algorithm to a neighborhood of the unknown optimum using a Lyapunov stability argument. While our dither is of low frequency, we have observed that the PE condition is satisfied after turning the dither on as shown by the following integral calculated over 1 second after turning on the algorithm

$$\int_{500s}^{501s} c(\tau)c^T(\tau)d\tau = \int_{500s}^{501s} = \begin{bmatrix} 2.0250e - 01 & 1.4631e - 03 \\ 1.4631e - 03 & 1.4856e - 05 \end{bmatrix} \geq (4.3e - 06) * I \tag{A10}$$

At this time, we do not have a full explanation for the rapid satisfaction of the PE condition.

It must be noted that six parameters are needed to define the PI-ESC algorithm: $k_p$, $\tau_I$ & $k_b$ for the PI control Eq. (2) and $K$, $k_T$, & $\sigma$ for parameter estimation. In this paper we have used trial and error to determine these parameters. The selection of these parameters was done using a clean blade (no contamination or erosion) and then fixed at these design values for the degraded blade cases, which would be a reasonable way of deploying the algorithm in the field. The parameters were designed assuming 8m/s mean wind speed, 10% TI. Parameter selection by trial and error requires initial conditions for the TSR other than the optimal value for clean blades $\lambda_{\text{opt}} = 7.6$. These initial conditions were taken at $\lambda = 9$ (above optimal) and $\lambda = 6$ (below optimal). Both the dither frequency and amplitude were determined following the procedure explained below.

To complete the PI-ESC design, the frequency and amplitude of the sinusoidal dither signal $d(t)$ in Eq. 2 must be specified. Again, assuming a clean blade operating at its optimum TSR, the dither frequency is chosen within the bandwidth of the plant dynamics as recommended in Ariyur and Krstic (2003); Rotea (2000). The rotor inertia and the actuator dynamics yield the input dynamics. The input dynamics is merged with the plant and estimated using open-loop step responses under constant wind input to simplify the design. The response of the rotor speed ($\omega_r$) under staircase step changes in the tip-speed ratio indicates a second-order dynamics (a first-order wind turbine dynamics and the dynamics of the generator torque PI controller, approximately), as shown in Figure A1. The top plot in Figure A1 shows the staircase tip-speed ratio command to the ROSCO and the OpenFAST tip-speed ratio output, while the rotor speed is shown in the bottom plot. The test was performed for the hub-height mean wind speed of 8 m/s with no turbulence. Based on the step response of the rotor speed, natural frequency ($\omega_n$) and damping ratio ($\zeta$) were calculated for each step change case. The undamped natural frequency $\omega_n$ ranged between 0.36 rad/s to 0.5 rad/s while $\zeta$ ranged between 0.56 to 0.77. Then we calculated the time constant (i.e., $\tau = \frac{1}{\zeta \omega_n}$) for each case and the slowest combination (i.e., largest time constant) was adopted for the LP-PIESC design,i.e., $\omega_n$=0.36 rad/s and $\zeta$=0.76. The corresponding bandwidth is 0.33 rad/s. Since dither frequency should be selected within the estimated bandwidth, it was selected conservatively as 0.16 rad/s. The Bode plot for the estimated plant dynamic is shown in Figure A2. The dither amplitude was selected using trial and error.

Finally, The parameters of the LP-PIESC scheme are listed in Table A1. The minimum and maximum saturation limits for the tip-speed ratio set-point were set at 4 and 10 (Eq. 2) to avoid very low or very high TSRs set point changes.

We conclude this appendix with further details concerning tuning the six PI-ESC parameters needed to define the algorithm: $k_p$, $\tau_I$ and $k_b$ for the PI control in Eq. 2 and $K$, $k_T$, and $\sigma$ for the estimator in Eq. A5. The trial and error process is a simulation-based method where we looked for consistent parameter convergence and $\hat{\theta}_1$ converging to zero, which we know is the correct asymptotic value for this parameter. Once these three parameters were tuned, we tuned the PI parameters for better TSR convergence to its known value for a clean blade. After the desired response was obtained (on a clean blade), parameters were fixed and used for all other simulations with the LP-PIESC and different mean wind speeds and turbulence intensities.

*Author contributions.* DK: Data curation, Formal analysis, Investigation, Software, Visualization, Writing - original draft preparation. MR: Conceptualization, Funding acquisition, Methodology, Resources, Supervision, Visualization, Writing - review & editing.

*Competing interests.* No competing interests are present.

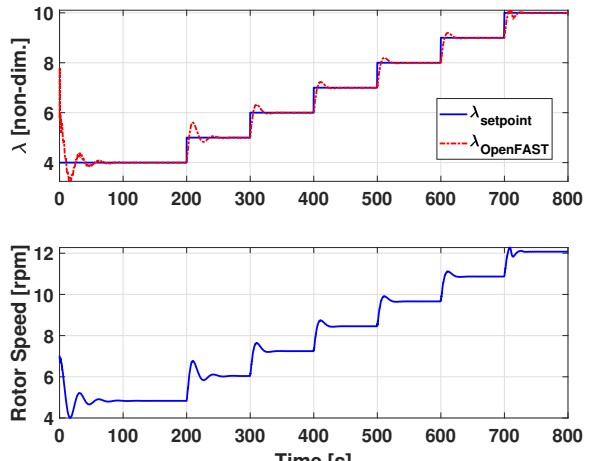

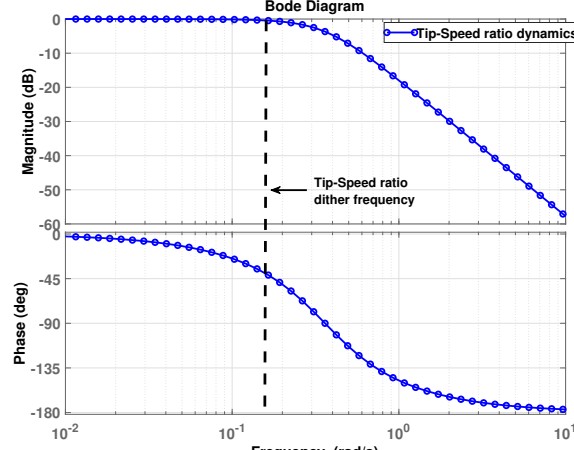

**Figure A1.** The top plot shows the staircase tip-speed ratio command to the ROSCO and the OpenFAST tip-speed ratio output while the bottom plot shows the rotor speed response.

**Figure A2.** Bode plot of plant dynamics, and the dither frequency.

**Table A1.** LP-PIESC Parameters for tip-speed ratio set-point adjustment (see Fig.4).

| Parameter | Tip-Speed Ratio LP-PIESC |
|---|---|
| Dither Frequency ($\omega$) | 0.16 rad/s |
| Dither Amplitude ($a$) | 0.1 (non-dim.) |
| $k_T$ | 25 rad/s |
| $K$ | 20 rad/s |
| $\sigma$ | $10^{-6}$ (s/rad)$^2$ |
| $k_p$ | 0.03 s/rad |
| $\tau_I$ | 2.1 (non-dim.) |
| $k_b$ | 1 rad/s |

*Acknowledgements.* This work was supported in part by the Center for Wind Energy at the University of Texas at Dallas.

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
