# Peer review of "Brief communication: Real-time estimation of optimal tip-speed ratio for controlling wind turbines with degraded blades"

_Wind Energy Science, 2023_

## Author Comment (AC1)

**Response to Reviewer 1**

Many thanks for your detailed comments and questions regarding our Brief Communication. In this response, we have included your comments/questions (framed, black font, italics) followed by our response (red font) to each point raised. When references are used, they can be found in the submitted manuscript or full citations are given in this response.

When preparing our response we noted a typo in Table 2 in the paper. The value of the proportional gain $k_p$ must be changed from 3e-6 s/rad to 3e-2 s/rad.

> *"This paper considers the optimization of wind turbine torque controllers that track an optimal TSR value for below-rated load operation. Due to wear/tear/ fouling the properties of the turbine rotor change over time, possibly leading to a shifted location of the optimal TSR, leading to suboptimal operation when this parameter remains nominal and uncalibrated. A novel LP-PIESC scheme is proposed to calibrate the TSR setpoint value.*
>
> *The motivation of the paper is relevant, although numerous other works have been done in this field, which should be included and acknowledged in the introduction. Furthermore, I think that the major contribution of this paper is the LP-PIESC scheme, which supposedly can provide faster (instantaneous?) convergence. First, the new ESC scheme is not well described, leaving many questions about its working principles. Second, the tuning procedure of the scheme is not explained (only Appendix A elaborates on the excitation frequency). Third, the instantaneous convergence results are questionable: there is no excitation before the convergence, so how is the gradient obtained? Also, please explain how it is possible that there is instantaneous convergence; to me, this seems impossible without prior knowledge of the optimal setpoint value. Overall, I think this work needs significant improvements before it can be considered for publication in WES."*

The aim of our work is to provide numerical evidence of the existence of a class of extremum seeking algorithms (LP-PIESC) that can quickly identify optimal TSR despite changes in average wind speed and turbulence intensity (TI) for situations where TSR values have shifted to unknown values. We heavily rely on existing literature on extremum seeking control without presenting rigorous mathematical proofs.

The LP-PIESC algorithm has two main components: an identifier for parameter estimation and a PI controller using one of the the estimated parameters. Both components have been described in great detail already; see, for example, Guay and Dochain (2017). The parameter estimation in this reference has roots in prior work such as Guay and Dochain (2015), "A time-varying extremum-seeking control approach," Automatica, 51. Thus, there is already available literature to understand the working principles of this algorithm. Note that our WES manuscript is a "Brief communication." Thus, we must rely on the open literature for existing work. Having said that, we are happy to provide some intuition behind the parameter estimation algorithm to facilitate understanding of the working principles of the PIESC. The intuitive arguments given below are based on the work of Guay, M., and Dochain, D. (2015).

Let us consider a simpler static optimization problem, where the goal is to minimize an instantaneous cost $J(u)$ by proper choice of control $u$. Let $y$ denote the instantaneous measurement of $J(u)$. That is, $y = J(u)$. Now take the time derivative of $y$ to get

$$\dot{y} = \frac{\partial J}{\partial u}^T \dot{u} = \phi^T \theta \tag{a}$$

where $\phi = \dot{u}$ is the regressor and $\theta = \partial J/\partial u$ the time varying gradient we seek to identify.

Let $\hat{\theta}$ denote the estimate of $\theta$ at time $t$. The algorithm to estimate $\hat{\theta}$ belongs to a class of prediction error methods for problems with time-varying parameters. The proposed predictor $\dot{\hat{y}}$ for the output dynamics $\dot{y}$ is given by

$$\dot{\hat{y}} = \phi^T \hat{\theta} + Ke + c^T \dot{\hat{\theta}} \tag{b}$$

where $e = y - \hat{y}$ is the output predication error ($y$ is measured), $K$ is a constant (scalar) gain to drive the prediction dynamics $\dot{\hat{y}}$ with the output prediction error $e$ and $c^T$ is a filtered regressor obtained by low pass

filtering $\phi$; i.e.,

$$\dot{c}^T = -Kc^T + \phi^T \tag{c}$$

where $K$ is the scalar gain introduced in (b). Note that $c^T\dot{\hat{\theta}}$ also drives the predictor dynamics $\dot{\hat{y}}$ to account for the time-varying nature of $\theta$.

The parameter estimate $\hat{\theta}$ we seek is calculated from the following differential equations:

$$\dot{\hat{\theta}} = \text{Proj}(\Sigma^{-1}(c(e - \hat{\eta}) - \sigma\hat{\theta}), \hat{\theta}) \tag{d}$$

$$\dot{\Sigma}^{-1} = -\Sigma^{-1}cc^T\Sigma^{-1} + k_T\Sigma^{-1} - \sigma\Sigma^{-2} \; (\dot{\Sigma} = cc^T - k_T\Sigma + \sigma I) \tag{e}$$

$$\dot{\hat{\eta}} = -K\hat{\eta} \tag{f}$$

Guay and Dochain (2015, 2017), define the auxiliary variable $\eta := e - c^T\tilde{\theta}$, where $\tilde{\theta} = \theta - \hat{\theta}$ is the actual parameter error (with $\theta$ unknown). Note that $c^T\tilde{\theta} = e - \eta$. Thus, if we also want to drive the update differential equation (d) with information about the time-varying parameter error $\tilde{\theta}$, then using $e - \hat{\eta}$ in (d) seems reasonable.This argument is based on intuitive interpretations of variables only.

A rigorous proof of convergence to a neighborhood of an optimum using this parameter estimation algortihm coupled with a specific extremum seeking control law for $u$ is given in Theorem 1 of Guay and Dochain (2015). It's proof is based on a Lyapunov argument. Let us now give an idea of the role of the matrix $\Sigma$ and the projection operator $\text{Proj}(\cdot)$.

The matrix $\Sigma^{-1}$ is the gain matrix for the parameter update law (d) This matrix has an update law (e) similar to the one in continuous-time least-squares with forgetting (Shaferman et al., European Jou. of Control, V. 62, Nov. 2021). In eqn. (d), the operator $\text{Proj}(\cdot)$ is a Lipschitz projection operator designed to ensure that the estimates are bounded within a fixed constraint set. This projection algorithm was implemented as discussed in Appendix E of (Krstic et al. (1995), "Nonlinear and Adaptive Control Design," 1st edition, John Wiley & Sons Inc.) and the constraint set adaptation was adopted as per Adetola and Guay (2011) ("Robust adaptive MPC for constrained uncertain nonlinear systems," Int. J. Adapt. Control Signal Process., 25, 155–167). Given the space limitations of a WES Brief Communication, it is not possible to provide all the equations. However, we could provide all the relevant references as done in this reply if a revised version is invited.

This approach from Guay and Dochain (2015) is then extended in Guay and Dochain (2017) using a PI controller for extremun seeking. This is the method we use in our Brief communication. In this case, there is dynamics between the control variable $u$ and the cost function we seek to optimize. Thus, there is an extra parameter to model the dynamics of the instantaneous cost function $y$, which is parameterized by the differential equation

$$\dot{y} = \theta_0 + \theta_1(u - \hat{u}) = [1 \;\; (u - \hat{u})] * [\theta_0 \;\; \theta_1]^T = \phi^T\theta \tag{g}$$

where $u$ and $\hat{u}$ are defined in equation (2) in our paper. Then we apply back calculation anti-windup to obtain $u^s$. This is done because we want to maintain the TSR within practical limits. The final TSR set-point $\lambda_{sp}$ is obtained by rate limiting $u^s$ to smooth the final set point. In equation (g) one may associate $\theta_0$ with the dynamics of the system we seek to optimize and $\theta_1$ with the gradient of the cost function. Note that now the parameter we seek to estimate is $\theta = [\theta_0 \;\; \theta_1]^T$ and the regressor is $\phi = [1 \;\; (u - \hat{u})]^T$. The remaining equations for the identifier are as shown in the text above and in Fig. 5 of the paper.

**In a revised version of this paper we could eliminate Fig. 5 and instead give a clear block diagram of the solution with input $P$ and output $\lambda_{sp}$ ($\lambda_{opt}$ in Fig. 1). Due to space limits we will also eliminate Fig. 1 and instead write down that our solution provides the TSR set-point value required by the NREL ROSCO controller. In addition, we could provide the equations (with brief explanation) for the parameter estimation algorithm in the text rather than inside the block diagram in Fig. 6.**

To conclude the response to your opening comments, time series of key signals are given to enhance the understanding the working principle of the algorithm. First note that the complete algorithm has been implemented in Simulink with a step size of 0.0125 sec for discretization (i.e., we have 80 Hz sampling frequency). In this explanation, we focus on the case of eroded blade, where the TSR set point needs to move from 7.6 (for clean blade) to 8.4 for the eroded blade; i.e., a 0.8 increase in TSR (see Fig. 4 in the paper). As shown in the block diagram of Fig. 5, $\hat{\theta}_1$ is the only estimated parameter used by the PI controller

in equation (2). The Simulink scope in Fig. A shows the time series of $\hat{\theta}_1$.[1] Only 1 sec of the simulation leading to Fig. 12 in the paper is shown, which represents about 80 samples. The two terms of PI controller are shown in Fig. A (right-hand side). Note that the proportional term reaches 0.8 in one sample after the LP-PIESC is turned on at 500 sec. Then the proportional term decays to zero and the integral term $\hat{u}$ grows to 8.4 asymptotically after multiple samples. The estimated new TSR is the sum of the proportional and integral terms. We rate limit the result to obtain the final TSR set-point to ROSCO as shown in Fig B. The final value is above 8.4 due to the addition of the sinusoidal dither. Note the accelerating effect of adding the proportional term (entry 1-2 in Fig. A) relative to using the integral term only (entry 2-2 in Fig. A). While we do not have a formal proof, this effect resembles the increase in bandwidth that a PI controller can offer over a P-only controller.

[Figure]

Figure A: Simulink Scope for LP-PIESC signals – Eroded blade (U=8m/s, TI =10%).

[Figure]

Figure B: Simulink Scope showing TSR command to the ROSCO controller – Eroded blade (U=8m/s, TI =10%).
* * *
[1]If one thinks of $\hat{\theta}_1$ as the relevant gradient, note that it is negative because the algorithm actually minimizes the sign inverted performance function. This can be done because the max of a function $f$ can be obtained by minimizing $-f$.

"- Announce in the abstract what type of wind turbine controller you are assuming.

- State the significance of using LP-PIESC as compared to regular ESC, as this is the main contribution of the paper."

- Please note that the type of controller used (ROSCO) is mentioned in the last sentence of the abstract. This turbine controller is as developed by Abbas et al. (2022), which is referenced in the technical brief.

- Your suggestion to "state the significance of using LP-PIESC as compared to regular ESC" can be found in Kumar and Rotea (2022) for a different torque and pitch turbine controller. This reference actually shows a comparison between LP-PIESC and the conventional LP-ESC. Thus, since our technical brief has no comparison, we would prefer to keep this statement in the Introduction, line 52.

Introduction:

- You write: "The LP-PIESC has been shown to be a faster variant of the traditional perturbation-based ESC (Kumar and Rotea, 2022)." This might be interpreted that you do not need perturbation using LP-PIESC. You would still need a perturbation to estimate the gradient, right?

- A more elaborate literature study and acknowledgment of the works in the field of setpoint/model/controller calibration should be included.

- The dither signal ensures the persistence of excitation (PE) condition in assumption 5 of Guay and Dochain (2017). This assumption is used to prove the convergence of the PIESC algorithm to a neighborhood of the unknown optimum using a Lyapunov stability argument. While our dither is of low frequency, the PE condition is satisfied after turning the dither on as shown by the following integral

$$\int_{500s}^{501s} c(\tau)c^T(\tau)d\tau = \int_{500s}^{501s} = \begin{bmatrix} 2.0250e-01 & 1.4631e-03 \\ 1.4631e-03 & 1.4856e-05 \end{bmatrix} \geq (4.3e-06)*I \qquad (h)$$

where $c$ is the filtered regressor.

- Please note that our paper is a Brief Communication; thus, an elaborate literature study and acknowledgment of the works in the field of setpoint/model/controller calibration would be difficult to include. This is particularly notable in our case because we already need to reduce the length of the original submission. We can, however, include a couple of the top most cited papers in TSR estimation if requested by the associate editor. Of course, we welcome any suggestions you may have concerning salient archived journal papers in this area.

Background:

- Is it a valid assumption to have a precise measurement of the rotor effective wind speed?

- Fig 1: In the figure you indicate that $\hat{v}$ comes from a wind speed estimator, while in the text you say something different.

- Fig 3: There seem to be few data points for Cd-curves. Can you increase the resolution?

- We assume you are referring to the statement after line 85 in the submitted manuscript, where we explain the use of the rotor disk average (RtVAvgxh) wind speed calculated by OpenFAST and not from the wind speed estimator inside ROSCO. We understand that the ROSCO wind speed estimator makes use of the power curve, which changes as the blades degrade. Thus, ROSCO may not calculate the

correct rotor speed reference from the TSR set-point unless the power curve is modified to account for degradation/erosion. While our assumption may not be most practical, it is necessary to demonstrate the LP-PIESC by avoiding unknown complexities of ROSCO's wind speed estimator (which is not the main purpose of our study).

- We introduced Fig. 1 (in paper) as a graphical representation of the ROSCO architecture, as per Abbas et al. (2022). The reason for using the rotor disk average (RtVAvgxh) wind speed has been explained already. To reduce the length of the Brief Communication, it is likely that we would need to remove Fig. 1 and add a few sentences with the key aspects of ROSCO used in our study.

- The resolution of lift and drag coefficients for the clean blade as well as the contaminated and eroded blade are shown with markers in Figure C below. The aerodynamic coefficients for clean blade were obtained from the NREL OpenFAST package of the 5MW reference turbine available at `https://github.com/OpenFAST/r-test/blob/main/glue-codes/openfast/5MW_Baseline/Airfoils/NACA64_A17.dat`. These were used to run simulations to obtain $C_p$ for every 0.2 change in TSR (shown in Fig. D). If this resolution of $C_p$ is considered low, we can increase it but we cannot increase the resolution of the lift and drag coefficients as those are the only points for which we have available data for the NACA64 airfoil. We are happy to replace the plots in the Brief Communication with the ones shown below.

[Figure]

[Figure]

Figure C: Change of lift and drag coefficients for NACA64 airfoil.

Figure D: Nominal and modified (due to contamination and erosion of blades) $C_P - \lambda$ curve for NREL 5-MW wind turbine reference model.
* * *
*Section 2*
*- You state that $\theta_1$ is proportional to the gradient. It could be better explained that in (2), you can observe that this quantity is subject to a proportional action with kp and an integral action with saturation capabilities.*
*- What estimation problem are the parameters in $\theta$ a result of? What do they represent, and in which context?*
*- The paper does not describe the rationale behind the gradient estimation scheme. As this is the major contribution of the paper, you should have a proper description of its working principles.*
* * *
- The statement we made on $\hat{\theta}_1$ is intuitive but not rigorous. Per equation (g), $\hat{\theta}_1$ estimates a parameter used to model the influence of the control increment (i.e., $u - \hat{u}$) on the dynamics of the cost function. From this equation, the "gradient interpretation" can be understood, at least intuitively.

- The actual estimation problem solved is for the 2D vector $\theta^T := [\theta_0, \theta_1]^T$ in equation (g). These parameters are used to parametrize the time derivative of the performance index $y$ (e.g., log of power after moving average filtering) using

$$\dot{y} = \theta_0 + \theta_1 (u - \hat{u}) \qquad (i)$$

which forms the basis for the parameter estimation algorithm in the box of Fig. 5. Key elements of this algorithm where intuitively explained at the beginning of this reply. Further details can be found in Guay and Dochain (2017).

- We believe the introductory part of this reply cover the essentials of the LP-PIESC algorithm. Space permitting, we would be happy to include the estimation problem being solved, with identifier equations, in a revised Brief communication if requested.
* * *
*Section 3*
*- Consider using a more state-of-the-art reference turbine model, like the IEA 15 MW turbine.*
*- Fig 5: The complete algorithm of finding $\theta$ is given in this figure in the large block, without any explanation. It is too complex to understand from a list of relations without explanation and justification!*
*- All Figures in the paper are given without a proper elaborate caption that allows for interpretation of the figure. Improve on this.*
*- Table 2: How did you arrive at these ESC parameters? Through trial and error or a systematic tuning procedure?*
*- In Appendix A, you provide justification for the dither frequency but not for the other values.*
*- You write: "The LP-PIESC converges to the new optimal tip-speed ratio almost instantaneously for all the cases." How is this possible? As far as I understand, you only estimate the gradient in the form of $\theta_1$. Instantaneous convergence is only possible if you know how far you are from the optimum value, e.g., tuning the proportional gain to precisely the correct value. But this is just guessing, and maybe I am missing something. However, the paper does not clarify this aspect.*
*- Figure 8: How can the gradient be estimated without perturbation before 500 s? How is it possible to arrive at the optimal value instantaneously?*
*- It is unclear which variable you excite by dithering, is this $\lambda_{sp}$?*
* * *
- This will be considered for future work. Currently, we do not have the resources to apply the LP-PIESC to the IEA 15 MW as the first author has moved to industry and this project has concluded.

- We agree with you that Fig. 5 is complex to understand. We will have one single figure with input $P$ and output $\lambda_{sp}$. The parameter estimation algorithm can be given as separate equations as done in (Kumar and Rotea, 2022), and brief intuition behind these equations provided. Given that this reply can be included with the paper, the beginning of this reply should help readers gain additional insight behind the method. Please note that we cannot include the rigorous proofs given in Guay and Dochain (2017), but we can cite them.

- We feel we already have well detailed captions for all figures. Please let us know if there are any specific figure captions that would need update.

- Table 2: yes, all parameters in Table 2, except the dither frequency, are obtained by trial and error at 8 m/s wind speed and 10% TI for the **clean blade** (see line 250). Parameter tuning is an area of improvement for this algorithm, which is left for future research. Please note that due to the use of the log-of-power, once we calibrate parameters at one wind condition, the same parameters can be used at other wind conditions. We have observed this behavior in simulations (this technical brief, Kumar and Rotea (2022) and in wind tunnel experiments (Kumar et al. (2023), "Wind plant power maximization via extremum seeking yaw control: A wind tunnel experiment," Wind Energy, Vol. 26 (3))

- In a revised version of the paper we will state that all parameters, other than dither frequency, have been obtained by trial an error.

- Please refer to Figs. A and B in this reply, which we believe clarifies the point you raised on convergence.

- As shown in Fig. A, the parameter $\theta_1$ ("the gradient") is being estimated after 500s. It takes around 0.4-0.5s for this parameter to converge to zero.

- The variable that is being excited is $u$ - see eqn. 2 - which is the set point $\lambda_{sp}$ for the TSR.

**Response to Reviewer 2**

Many thanks for your detailed comments and questions regarding our Brief Communication. In this response, we have included your comments/questions (framed, black font, italics) followed by our response (red font) to each point raised. When references are used, they can be found in the submitted manuscript or full citations are given in this response.

Before responding to your points, please note that in the response to reviewer #1, we provided explanations on the working principle of the LP-PIESC algorithm. There are several details, provided through "the lens of continuous time algorithms for estimating time-varying parameters," which is our case. We hope that the response to reviewer #1 plus specific answers to your points are satisfactory.

In addition, please note that there is a typo in Table 2 of the paper. The proportional gain $k_p$ should be replaced by 3e-2 s/rad.

> *This paper presents an extremum seeking controller for optimizing a wind turbine controller's tip speed ratio set point.  It's nice that the control scheme fits with an existing wind turbine architecture.  However, the benefit of using the log of the power is unclear, and the algorithm seems to converge to the optimal solution too quickly without adequate explanation.*

Our algorithm uses the log-of-power signal to determine the optimal TSR set-point. The significant advantage is that ln(P) = constant + ln(Cp) + 3*ln(V). Thus, for algorithms based on some form of gradient estimation, like ours, the gradient of log(P) depends only on Cp, which is exactly the quantity we seek to maximize. This use of the "log" transformation has huge implications because it eliminates the need to schedule control parameters on wind speed. That is, once you calibrate the any gradient-based algorithm for one wind speed, the same parameter values work at other wind speeds. Given that between cut-in and rated wind speeds, V^3 could change significantly, the use of log-of-power is highly advisable when adjusting control parameters (e.g., TSR, Torque Gain, Pitch Angle) to improve performance. The use and advantages of log-of-power have been explained and documented earlier in Rotea (2017) and Ciri, et al. (2019). In addition, we are providing two charts below that articulate this point.

[Figure]

[Figure]

The issue of quick convergence is addressed in the response to your major questions below and the reply to reviewer #1.

> *How does the algorithm converge to the optimal solution before a single dither signal cycle can compute the gradient? I think that justification, in wind energy terms, should be provided in this article.*

We do not have rigorous proof of rapid convergence for this specific application of LP-PIESC. Having said that, please see the response to reviewer #1, where we provide empirical evidence of convergence using a persistence of excitation condition like the one required in the main reference of our work Guay and Dochain (2017; eqn. (22)). The method we use for parameter estimation is not the conventional perturbation/demodulation method used in prior ESC algorithms, which is slower and

may require a few full dither cycles to estimate unknown parameters. Instead, we use a method that is applicable to estimating and tracking time-varying parameters. This method, developed by Guay and co-authors, has similarities with continuous-time least-squares with forgetting (Shaferman et al., European Jou. Of Control, V. 62, Nov. 2021), which may not require a full cycle of the dither to converge if a persistence of excitation (PE) condition is met. This PE condition involves the positive definitiveness of a matrix obtained by integrating $c(t) * c(t)^T$, where $c(t)$ is the filtered regressor defined by equation (c) in the reply to reviewer #1, with $\phi^T = [1 \quad u(t) - \hat{u}(t)]$ denoting the actual regressor (see eqn. (g) in the reply to reviewer #1), where $u(t)$ is the commanded TSR with no anti-windup compensation.

While we do not have formal proof, we believe that when the LP-PIESC algorithm is turned on, and the power (i.e., the input to the LP-PIESC) is fluctuating, the PE condition is met rather quickly, which might explain the rapid convergence of the parameter $\widehat{\theta_1}$. See Figs. A and B and eqn. (h) in reply to reviewer #1. In addition, please note that the Cp curves between 7.6 TSR (initial condition for the algorithm) and the new optimal TSR values shown in Fig. 4 in the paper are fairly simple concave segments with well-behaved slopes, which combined with the small TSR increase required to reach optimal values, suggest that rapid convergence may not unrealistic.

> *How exactly is the gradient estimated over time? What signals from the turbine are needed? The variables in Fig. 5 are not defined in the text. Can you show the gradient estimate over time?*

The only signal we need from the turbine is the rotor power $P$ (not the aerodynamic power), which is correlated with the turbine's electrical power output. The instantaneous rotor power is time-averaged with a moving average filter and then the natural log is applied. Fig. 5 is unclear and not consistent with Fig. 6. This situation will be corrected in a revised version of the paper. Now, with some abuse of notation, if we let $y(t) = \ln P(t)$, then this is the input signal to the PI-ESC algorithm in Fig. 5 (note that $y$ is misplaced in the diagram; it should be after the natural log block).

The equations for estimating the parameter $\widehat{\theta_1}$, which can be thought as a gradient, are given in pages 1-3 in the reply to reviewer #1. The time series of $\widehat{\theta_1}$ is shown in Fig. A (reviewer's #1 reply) for the case of eroded blade.

> *It appears that the TSR set point reaches the "optimal" before the power coefficient or actual tip speed ratio changes in any measurable way. How is this possible? The bandwidth of the torque controller limits the actual TSR; how can this algorithm converge faster than the torque controller?*

A change in the TSR or Cp may not be necessary for the algorithm to converge. As mentioned earlier, and in the response to reviewer #1, $\widehat{\theta_1}$ is the key parameter that needs to converge to determine the new optimal TSR. This parameter is used to parametrize the time derivative of the performance index (log-of-power in our case); see equation (g) in response to reviewer #1 and eqn. (8) and in Guay and Dochain (2017). While we do not have formal proof, we believe that when the LP-PIESC algorithm is turned on, and the power (i.e., the input to the LP-PIESC) is fluctuating, the PE condition is met rather quickly, which might explain the rapid convergence of the parameter $\widehat{\theta_1}$. Please note that all simulations in OpenFAST are sampling signals at 80 Hz (0.0125 sec sampling interval). In addition, the accelerating effect of the proportional term in the controller contributes to the fast convergence to the new optimal value for the TSR. One may see this effect by adding the (1,2) entry and the (2,2) entry in Fig. A in the reply to reviewer #1. The effect of adding the proportional term is like the increase in control bandwidth obtained when replacing a pure integral controller (as in several prior ESC algorithms) with a PI control law.

> *From cited work within this article, the authors claim that the log of the power allows the Cp to be maximized directly without requiring the wind speed. \frac{\del J}{\del u} = 1/Cp \frac{\del Cp}{\del u}. Doesn't the Cp in the denominator depend on the wind speed?*

The known approximations of Cp show that the power coefficient is a strong function of TSR and blade pitch angle but not necessarily the incoming mean wind speed V in isolation. See, for example, Carpintero-Renteria *et al.*, "Wind turbine power coefficient models based on neural networks and polynomial fitting," IET Renewable Power Generation, Vol. 14, Issue11, August 2020, which contains a complete review of existing Cp models. As explained earlier in this reply, the main advantage of using log-of-power is the removal of $V^3$ from the gradient of the performance index we seek to maximize. Please refer to Rotea (2017) and Ciri, et al. (2019) to see how calibration of parameters (these papers do not use LP-PIESC) at one wind speed $V$ works at any other wind speed below rated without the need for retuning algorithm parameters – this is a significant benefit of using the logarithm before processing the power signal.

> *In Fig. 8, there is a step change as soon as the algorithm is enabled, and then it seems to converge slowly to another point. How do you account for this behavior? Was an initial guess provided to the algorithm?*

The initial TSR was set at 7.6 (i.e., start with the clean blade optimum). To answer your question, we run one of the simulations in Fig. 8 for 3000 s, instead of the 1500 s in the paper. The result is shown below in Fig. A2 below. Note that the dither is active since the algorithm is turned on at 500 s. The time series of the TSR set point after 1500 s

appears to oscillate (dither with amplitude 0.1 or 0.2 peak-to-peak around a mean value between 8.1 and 8.2 (8.2 is the optimum for this case, as it can be seen from Fig. 4). Note also from Fig. 4 (and Fig. D in reply to reviewer #1) that Cp does not change much (less than 10%) between TSR 8.1 and TSR 8.2 for the case of contaminated blade.

[Figure]

Figure A2: Tip Speed Ratio setpoint (LP-PIESC output) – Contaminated blade (U=9m/s, TI =10%).

Please note that the dynamics of the turbine with ROSCO and LP-PIESC is complex. For the simulation shown in Fig. A2, there are time intervals where the turbine is in above-rated conditions (see pitch signal in Fig. 9 of the paper). Thus, any small fluctuations around the mean value of the commanded TSR, which are not accounted for by dither, would require an analytical investigation with the full nonlinear system. Our paper is only aimed at providing numerical evidence that the LP-PIESC offers an alternative to retune the TSR. In a practical application, one could turn off the dither as soon as the new optimal TSR is reached. A stopping criterion for the dither could be based on the magnitude of the parameter $\widehat{\theta_1}$ (see Fig. A in reply to reviewer #1). In this paper, we have run the dither continuously.

---

## Author Response (AR1)

**Authors second response to reviewers**
**February 27, 2024**

On January 19, the authors provided a detailed response (12 PDF pages) to the reviewers' comments and questions. The response is archived as AC1 on the discussion tab of the paper's webpage entry. At that time no revised paper was uploaded to the journal's website. This was done assuming guidance, from reviewers and associate editor, would be provided on ways to address some of the points raised without extending too far beyond the typical length of a Brief communication. Note that while the paper was submitted as a Research article, it was re-classified as Brief communication by WES prior to review, which is acceptable to the authors.

The revised paper is now submitted. We believe this version addresses the main points raised by the reviewers. In particular,  we have added an appendix with the key equations of the parameter estimation method used by LP-PIESC. This revised manuscript provides insight into the method and arguments to support the faster convergence of the LP-PIESC relative to more conventional extremum seeking control algorithms using dither perturbation and demodulation to extract the necessary gradient information. This insight is backed up by citing the original papers on PI-ESC and other supporting publications.

The revisions to the paper are all in blue font. The initial response to reviewers (AC1, January 19) has a point-by-point response to the reviewers' comments and questions.

Please note that the initial response AC1, submitted on January 19, has some technical material (including graphs) that is meant to answer reviewers' questions but cannot be included in the Brief communication due to space limitations.

Sincerely,

Devesh Kumar and Mario Rotea

---

## Author Response (AR3)

| To: | Referee #3 |
|---|---|
| Subject: | Detailed response to suggestions for revision or reasons for rejection |
| From: | The authors |
| Date: | August 16, 2024 |

Many thanks for your detailed comments and questions regarding our Brief Communication. In this response, we have included your comments/questions (black font, italics) followed by our response (red font). When references are used, they can be found in the submitted revised manuscript or full citations are given in this response.

Please note that on January 19, 2024, the authors provided a detailed response (12 PDF pages) to the reviewers' comments and questions. The response is archived as AC1 on the discussion tab of the paper's webpage entry. Note that while the paper was submitted as a Research article, it was re-classified as Brief communication by WES prior to review, which limits the number of pages we can use. Whenever a point you raised has significant overlap with the January 19 response to reviewers #1 and #2, we indicate so by citing Author's Response AC1. Note also that on February 29, 2024, a revised paper version (with tracked changes in blue font) was uploaded after major revision decision from the first round of reviews. Currently, this first revised paper is not publicly available in the discussion section of WES, but we assume this version is the one you've reviewed.

*Reviewer: "The paper presents an extremum-seeking control (ESC) method to estimate the optimal tip-speed ratio. The application is well-suited for degraded blades. The topic is interesting and relevant for readers of Wind Energy Science. The paper is clear and well-written. I have some comments regarding the methodology. Typically in below-rated wind region, the K-omega2 law is used to maximise the power. Please see ([1], Section 3.1). In the paper, a PI tip-speed ratio/rotor speed set-point tracking controller is used, where the rotor speed set-point is computed by the estimated tip-speed ratio by ESC and the rotor-averaged wind speed. There are shortcomings and contradictions with this method, particularly with the assumption of a degraded blade."*

Authors: Thank you for noticing the clarity and well-written exposition of our paper. We agree with you that the K-omega2 law is an appropriate control law to maximize power. For this reason, we have first published the use of LP-PIESC for tunning the torque gain parameter K in Kumar and Rotea (2022; DOI: 10.3390/en15031004). The purpose of the current Brief was to demonstrate that LP-PIESC can also be used to estimate other controller parameters (in this case we chose TSR set point) besides the torque gain parameter K.

*Reviewer: "My concern is regarding the rotor-averaged wind speed, which could be obtained from an estimator, as suggested in the paper. But with the assumption of blade degradation, the aerodynamic properties (Cp surface) in the estimator model would be different to the ground truth. Thus, the estimated wind speed would be inaccurate. See [2]. Even though the tip-speed ratio can be accurately estimated by ESC. There is still a need to correct the estimator model, taking into account the degraded blade aerodynamics, to provide an accurate wind speed. This shortcoming wouldn't exist if the K-omega2 law was used in the first place. Perhaps the authors could include some discussions regarding the choice of K-omega2 law and tip-speed ratio tracking torque controller."*

Authors: Your concern is warranted. There is no point in identifying the correct TSR set point if the estimated wind speed used to compute the actual generator speed set point is wrong. This would be the case if any wind speed estimator/observer uses the power coefficient without any correction for blade degradation. This is why, as stated in line 85 of the Brief communication we said: "*In our simulations, ROSCO takes wind speed estimate \hat{v} from the rotor disk average (RtVAvgxh) calculated by OpenFAST.*" While this is not practical in a real application, we did this to eliminate the use of any observer parameters that could change with blade degradation. In this way, we can demonstrate how LP-PIESC can identify the correct TSR in an ideal scenario. Due to space limitations, we are not able to expand on [2] and other recent references that have the potential to circumvent this problem using machine learning or sensors. A statement to clarify this point has been added in red font after line 85, which also addresses your question "*on how to obtain the wind speed estimate in real life, especially with a degraded blade.*"

*Reviewer: Other comments:*

*1. "Literature studies on extremum seeking control, especially its application in wind energy, are lacking. I think it is important as this is a paper about using ESC. For example, [3] is one of the earliest used ESCs in turbine control. The study [4] used ESC in a large-eddy simulations. [5] performed a full-scale test with ESC. [6] proposed LiDAR-assisted ESC without knowledge of optimal tip-speed ratio."*

Authors: Please note that [3] was already cited in the Brief. We have included (see additions in red font) [3] and other existing references in the introduction as well as [4,5,6].

*2. "Page 3. In this architecture, both the generator torque ($\tau g$) and the blade pitch angle ($\beta$) are governed by PI controllers." As discussed above, please elaborate on why the K-omega law was not used."*

Authors: As mentioned above, LP-PIESC for the K-omega^2 law is already published in Kumar and Rotea (2022; DOI: 10.3390/en15031004).

3. *"Page 3. "In our simulations, ROSCO takes wind speed estimate ˆv from the rotor disk average (RtVAvgxh) calculated by OpenFAST." Based on the discussion above, please comment on how to obtain the wind speed estimate in real life, especially with a degraded blade."*

Authors: A practical (but potentially expensive option) would be to use a LIDAR as done in [6], which is a reference you provided. Alternatives based on machine learning techniques (such as [2]) are being proposed also but their practical application is not known to the authors. Please see the statement added in red font after line 90 of the second revision.

4. *"Page 6, Line 140. Please add some text to describe section 3 and section 3.1."*

Authors: Thank you for the suggestion. Additional text has been added in red font.

5. *"Figure 6. I can see there is a sine wave on the tip-speed ratio, which is caused by the dither function in the ESC? If so, the generator torque tracking a sinusoidal tip-speed ratio will also exhibit the sinusoidal behaviour? Can you comment on that? Maybe with a frequency spectrum of the generator torque."*

Authors: The time series of the generator torque is shown in Fig. 7 of the paper for the contaminated blade case. The spectrum of this signal after LP-PIESC is turned on (500 s) is shown below. The peak at the dither frequency (0.025 Hz) is clearly seen, but it is not the largest peak. This peak can be eliminated if the dither is turned off with an appropriate stopping criterion. We have not done this in the present paper; incorporating an appropriate stopping criterion is left for future work. Due to space limitations, we are not including the spectrum in the paper, but added a brief statement (red font) in the conclusions section of the revised paper.

References

[1] Abbas, N. J., Zalkind, D. S., Pao, L., & Wright, A. (2022). A reference open-source controller for fixed and floating offshore wind turbines, Wind Energy Science, 7, 53–73, https://doi.org/10.5194/wes-7-53-2022

[2] Lio, W. H., Li, A., & Meng, F. (2021). Real-time rotor effective wind speed estimation using Gaussian process regression and Kalman filtering. Renewable Energy, 169, 670–686. https://doi.org/10.1016/j.renene.2021.01.040

[3] Creaby, J., Li, Y., & Seem, J. E. (2009). Maximizing wind turbine energy capture using multivariable extremum seeking control. Wind Engineering, 33(4), 361–388. https://doi.org/10.1260/030952409789685753

[4] Ciri, U., Rotea, M., Santoni, C., & Leonardi, S. (2017). Large-eddy simulations with extremum-seeking control for individual wind turbine power optimization. Wind Energy, 20(9), 1617–1634. https://doi.org/10.1002/we.2112

[5] Xiao, Y., Li, Y., & Rotea, M. A. (2019). CART3 Field Tests for Wind Turbine Region-2 Operation with Extremum Seeking Controllers. IEEE Transactions on Control Systems Technology, 27(4), 1744–1752. https://doi.org/10.1109/TCST.2018.2825450

[6] Meng, F., Lio, W. H., & Larsen, G. Chr. (2022). Wind turbine LIDAR-assisted control: Power improvement, wind coherence and loads reduction. Journal of Physics: Conference Series, 2265(2), 022060. https://doi.org/10.1088/1742-6596/2265/2/022060

[Figure]

| To: | Referee #4 |
|---|---|
| Subject: | Detailed response to suggestions for revision or reasons for rejection |
| From: | The authors |
| Date: | August 14, 2024 |

Many thanks for your detailed comments and questions regarding our Brief Communication. In this response, we have included your comments/questions (black font, italics) followed by our response (red font). When references are used, they can be found in the submitted revised manuscript or full citations are given in this response.

Please note that on January 19, 2024, the authors provided a detailed response (12 PDF pages) to the reviewers' comments and questions. The response is archived as AC1 on the discussion tab of the paper's webpage entry. Note that while the paper was submitted as a Research article, it was re-classified as Brief communication by WES prior to review, which limits the number of pages we can use. Whenever a point you raised has significant overlap with the January 19 response to reviewers #1 and #2, we indicate so by citing Author's Response AC1. Note also that on February 29, 2024, a revised paper version (with tracked changes in blue font) was uploaded after major revision decision from the first round of reviews. Currently, this first revised paper is not publicly available in the discussion section of WES.

Some of your comments below (including references to Figures) lead the authors to believe that your review makes reference to issues in the original manuscript, which is publicly available in the discussion section as https://doi.org/10.5194/wes-2023-144. As stated in the above paragraph, the first **revised** version (dated February 29 by the publisher) of the paper is not currently publicly available, but we assume you have access to it and cite it below in our response to your comments.

*Reviewer: "This paper presented a wind turbine torque controller that can track an optimal/suboptimal TSR value for below-rated region due to the change of the blade aerodynamic properties over time. A novel LP-PIESC scheme is proposed to calibrate the TSR set point value.*

*There are numerous other works have been done in this field. For example, there is one work here (https://dx.doi.org/10.1088/1742-6596/2265/2/022060), and another work investigated the wind speed estimation, which should be considered as a fundamental*

*base in order to make the LP-PIESC algorithm work. (10.1088/1742-6596/75/1/012082). Those previous studies should be included in the literature study part."*

Authors: The first reference is relevant and has been added as suggested also by reviewer 3. The second reference uses the power coefficient Cp to obtain the wind speed, which is not desirable in our context due to the changes in Cp under blade degradation. Blade degradation can also affect the parameters used in the observer to estimate aerodynamic torque. (See section 3 of 10.1088/1742-6596/75/1/012082 for further details*)*

*Reviewer: "Furthermore, I think that the major contribution of this paper is the LP-PIESC scheme, which supposedly can provide faster convergence.*

*I have the following general comments:*

1. *The LP-PIESC scheme is not well described;*
2. *The tuning procedure of the scheme is not well explained;*
3. *How is the gradient obtained?*
4. *How will the novel LP-PIESC work without the wind speed estimation or, as the author mentioned, without the pre-knowledge of physical model?"*

Authors: We believe that these points were already addressed in the first revision of the paper dated February 29. A description of the LP-PIESC was provided in section 2.3 and appendix A. The tunning procedure is simulation based; this was explained also. However, given your comment in item 2, we provide more details about tunning at the end of appendix A. Point number 3 has already been addressed in the first revision of the paper – see appendix A.

Item 4 is important. There is no point in identifying the correct TSR set point if the estimated wind speed used to compute the actual generator speed set point is wrong. This would be the case if any wind speed estimator/observer uses the power coefficient without any correction for blade degradation. This is why, as stated in line 85 of the Brief communication we said: "*In our simulations, ROSCO takes wind speed estimate \hat{v} from the rotor disk average (RtVAvgxh) calculated by OpenFAST.*" While this is not practical in a real application, we did this to eliminate the use of any observer parameters that could change with blade degradation. In this way, we can demonstrate how LP-PIESC can identify the correct TSR in an ideal scenario. Due to space limitations, we are not able to expand on this issue, but we recognize that the use of sensors (e.g., LIDAR) or recent advances in machine learning could produce accurate estimates of wind speeds without using model parameters that can change with blade degradation. A statement to clarify this point has been added in red font after line 90 of the second revision. Note that our emphasis is providing empirical evidence to support the claim that LP-PIESC can find important turbine

parameters  such as optimal TSR or the torque gain in the k-omega^2 control law (see https://www.mdpi.com/1996-1073/15/3/1004). Such evidence justifies continued research on the use of ESC-like methods to adapt turbine parameters when changes happen.

"Reviewer: Some more detailed comments:"

• *"In Fig. 8, the LP-PIESC controller is turned on at 500s, and then it seems to converge to a value of tip-speed-ratio not the same as the TSR set point. What does this happen? Can you explain this?"*

Authors: For the reasons given in this reply, the authors do not know which Figure 8 you are referring to (original manuscript or revised version 1 from February 29). Having said that, none of the figures show lack of convergence when the turbine is below rated wind speeds. The drops you see in the actual TSR (and Cp) occur at wind speeds above rated. In this case, the ROSCO activates the pitch controller as shown in the paper (Figure 7 of second revision) and also in the Figure below.

[Figure]

• *"Figure 7 only shows the time series of wind speeds at 7, 8 and 9 with TI = 10%, and it does not bring more added value to the paper. So please consider removing it."*

Authors: This is not correct. Figure 7 (first revision) shows key turbine time series: Generator speed, Generator Torque, Power and Blade Pitch time series. The original manuscript does have wind speeds in former Figure 7, but the original manuscript is not the one that should be reviewed in the second round. Your review is a second round and should be based on the revised manuscript we submitted on February 29, 2024.

• *The LP-PIESC algorithm is not described clearly even with the help of Figure 5, it is not clear how the gradient estimation is performed.*

Authors: Again, you are referring to the original manuscript. Figure 5 with the block diagram of the LP-PIESC algorithm has been deleted in the revised manuscript submitted on February 29. The algorithm description is in section 2.3 and appendix A of the revised version. Presumably you also read the author's response AC1 at https://doi.org/10.5194/wes-2023-144-AC1.

• *Some of the symbols in the Fig. 5 are not explained, which raises many questions to the reader/reviewer to accept the theory soundness of this algorithm. For example, what does $\theta$ represent? This is an important parameter of your algorithm. Please explain this*

Authors: Again, your comment applies to the original manuscript, which is the wrong version to review in the second round. The block diagram of the LP-PIESC was removed after the first review, and replaced with equations and explanations in the main text and the appendix.

• *Section 2, the author mentioned "The algorithm is gradient-based, which can adjust the tunable parameters to maximize a system's performance index in real-time without any physical models." I think this is not true. The ROSCO controller, which the author couples to, contains the wind speed estimation, which requires the pre-knowledge of physical models. Please explain this."*

Authors: We have addressed this comment already. See the response to your second comment, item #4. See also also the author's response to first round of reviews AC1 at https://doi.org/10.5194/wes-2023-144-AC1